# Investigating the impact of the years of blindness on sleep rhythms, dream patterns, and spatial abilities: The BLINDREAM protocol

Helene Vitali[1]*, Claudio Campus[1], Laura Giorgetti[2,3], Francesco Famà[2,3], Pietro Mattioli[2,3], Dario Arnaldi[2,3], Monica Gori[1]

**1** Unit for Visually Impaired People, Istituto Italiano di Tecnologia, Genova, Italy, **2** Clinical Neurophysiology, IRCCS Ospedale Policlinico San Martino, Genova, Italy, **3** DINOGMI, University of Genova, Genova, Italy

☙These authors contributed equally to this work.
* Helene.Vitali@iit.it

## Abstract

Sleep health is a topic of great interest in recent years. However, it is still often under-valued in vulnerable populations, such as those with sensory disorders. A higher percentage of blind individuals experience circadian disorders, and their dreams have different sensory contents. However, many questions remain unanswered regarding the impact of blindness on overall sleep structure and how circadian desynchronization may influence them. Additionally, understanding how changes in the dream sensory contents and sleep alterations may affect daily life skills, such as spatial cognition which tends to be more challenging in blind individuals, is relevant for further comprehension of the vision role on our perception. The BLINDREAM protocol aims to investigate the interrelationships of all these aspects. We aim to collect one week of data from 20 blind adults and 20 sighted age-matched controls. Participants' sleep and circadian rhythm will be assessed through one-night home polysomnography, melatonin sampling, one-week actigraphy monitoring, and questionnaires. Dream activity will be measured through a one-week voice-recorded dream diary and questionnaires. Finally, a neuropsychological assessment of spatial cognition will be conducted. Capturing the dynamics of subjective experiences phase-locked to neural and physiological evaluation both between and within individuals, this approach can contribute to our understanding of the impact of blindness on sleep processes.

## Introduction

The study of sleep has gained growing attention in recent years due to increased awareness among the general population. However, its relationship to sensory processing and how sleep processes are affected in sensory disabilities still remain intriguing questions. Blindness, in particular, is a great model to study the role of

**Data availability statement:** No datasets were generated or analysed during the current study. All relevant data from this study will be made available upon study completion.

**Funding:** This research was partially supported by the European Research Council (PI Monica Gori; Grant agreement No. 948349). https://erc.europa.eu/homepage No the funder plays any role in the study design, data collection and analysis, decision to publish, or preparation of the manuscript.

**Competing interests:** The authors have declared that no competing interests exist.

vision on sleep processes. The lack of vision poses unique challenges to sleep regulation, primarily due to the disruption of the circadian process, which is regulated by the light-dark cycle [1]. This disruption can lead to sleep phase disorders, most notably Non-24-Hour Sleep-Wake Disorder (N24SWD), a condition that affects approximately 72.2% of blind individuals [2,3]. N24SWD results from a misalignment between the internal circadian clock and the 24-hour day, causing significant difficulties in maintaining regular sleep patterns and impacting daily life [4]. Despite the known association between vision loss and circadian rhythm disturbances, the relationship between the years of blindness and circadian desynchronization remains insufficiently studied. Furthermore, while melatonin is significantly influenced by light exposure, its secretion patterns in blind individuals vary widely, ranging from similar to sighted individuals to highly disorganized [5], depending on the degree of vision loss [6].

The circadian rhythm disruptions could also affect the organization of the sleep stages. The impact of blindness on th sleep stages is still a topic of debate, with some studies suggesting alterations in sleep stages and microstructure [7–11], potentially due to the plastic changes in brain structures induced by visual deprivation, while other studies report no significant changes [12,13]. Specifically, reduced delta activity and slow-wave sleep (SWS) have been reported in blind individuals compared to sighted controls [7,9,14,15], though other studies found no such differences [12,13,16]. Findings on sleep spindles are also inconsistent, with reports of reductions [8], increases [10], or no change [12]. REM sleep in blind individuals seems to be characterized by longer but less frequent episodes, with reduced or absent rapid eye movements [7,11]. Altered REM features, such as the alpha-like rhythm and sawtooth waves, have been reported [7,11], though these findings are not consistently replicated [12]. Altogether, the current literature suggests a lack of consensus also due to the limited number of studies on this topic. Moreover, it remains unclear whether these changes are a direct result of blindness or a consequence of circadian rhythm desynchronization [16]. While accounting for circadian misalignment in analyses may appear to eliminate macro- and microstructural alterations, differences in underlying thalamo-cortical and cortico-cortical networks could still be reflected in sleep patterns [17]. Indeed, even in childhood, before circadian disorders typically manifest, blind children report more sleep complaints and already exhibit an alteration of the sleep macro and microstructure [13,18–24].

These disruptions not only affect sleep architecture but should also have a broader impact on daily life activity [25–27]. However, only one study has evaluated the effects of sleep phase disorders on alertness, mood, and task performance in blind adults [28]. Specifically, Lockley and collaborators found that blind individuals with normal circadian timing showed a typical pattern of declining alertness, mood, and performance as their time awake increased, as is commonly observed in sighted individuals. In contrast, blind individuals with non-entrained circadian rhythm exhibited impaired waking function during the day, influenced more by the timing within their internal circadian cycle than time awake. Given the difficulties that blind individuals experience in spatial perception and cognition [29–37], and the role of sleep

in processing and consolidating spatial information (e.g., [38–42], it is possible that disruptions in sleep mechanisms could further exacerbate these challenges. Indeed, on one side, numerous studies have highlighted the role of sleep and dreams in processing and consolidating spatial information (e.g., [38–42]), and insufficient sleep impacts the ability to perceive the space around us [43]. On the other side, blind individuals often struggle with both perceptual and memory-based spatial tasks [29,36,37,44], particularly under high cognitive load conditions. Scientific research on multisensory integration shows that vision provides the most precise and reliable information about the spatial properties of the external world, thus having a dominant influence on spatial perception [45,46]. Indeed, vision offers an immediate and detailed view of the environment with high spatial resolution [47], making it crucial for aligning neural representations of space across different senses [46,48]. The dominance of vision in spatial reasoning becomes especially clear when it interacts with other senses. For instance, during sensory conflicts, auditory and tactile perceptions are strongly shaped by visual information presented at the same time [49–52]. Therefore, the absence of visual experience from birth alters the ability to perform complex spatial tasks [29–37]. However, most of these spatial difficulties do not manifest in late blind individuals [33,53–55], suggesting that the first years of life can constitute a sensitive period for the development of spatial metrics. Nevertheless, as the years of blindness increase, spatial challenges grow, becoming similar between early and late blind individuals after about 20 years [56].

Despite the strong connection between these aspects, nowadays, little is known about how blindness might influence the relationship between sleep, including dreams, and spatial cognition.

The content of dreams is also influenced by blindness [57,58]. For example, visual content is typically absent in the dreams of congenitally blind individuals or those who lost their sight early in life [59–61], whereas auditory, tactile, gustatory, and olfactory elements are more prominent [60,62]. Interestingly, those with late-onset blindness who had some visual experience earlier in life report a higher percentage of visual content in their dreams, though still less than sighted individuals [59,60]. The relationship between the onset of blindness and the presence of visual dream content suggests an inverse correlation, with earlier onset leading to less visual imagery in dreams [59,60]. The impact of these different dream experiences on spatial cognitive functions, is an area of growing interest [58,62]. Can the different processing of dream content influence how sensory information is processed during our waking experiences, significantly affecting long-term performance, such as spatial abilities? Current theories suggest that dreaming is not merely a reflection of past experiences, but part of an active process where the brain elaborates and strengthens memory traces during sleep [63,64]. Therefore, waking experiences can be incorporated into dream content and reprocessed. Moreover, some studies have shown that sensory stimulation during sleep can alter dream content and influence waking performance, indicating a possible bidirectional link between dreaming and cognitive function [64–67]. This is also evident for spatial performance [42,68]. In this framework, dreams may serve as a form of mental simulation, integrating sensory input and contributing to the development or reinforcement of skills used in daily life. Building on this idea, it is possible that the visual dominance that characterized typical dreams might be linked to better spatial performance. This could be due to a higher level of processing of visual information during sleep, which may reinforce spatial abilities during wakefulness. But what happens in blind individuals? How spatial performance manifests differently in those who experience visual content in their dreams compared to those who do not? The study presented with this protocol aims to address these gaps by investigating the impact of blindness on sleep processes, dream content, and their relationship with spatial perception and memory. By exploring the links between clinical, psychological, neurobiological, and electrophysiological measures, this research seeks to provide a comprehensive understanding of how blindness influences these interconnected domains.

## Materials and methods

### Participants

We are aiming to recruit 40 adults: 20 Blind/Severely Visually Impaired (BSI) individuals and 20 age- and biological sex-matched Sighted (S) controls.

## Sample size

Given the proof-of-concept nature of the study and the unavailability of prior effect size estimates in the relationship between the different measures, calculating power is currently not feasible. Thus, the sample size is based on a provisional and ambitious estimate of recruitment capacity using the *pwr* package on the R environment [69]. However, previous literature studies compared between groups some of the measures that we are considering separately, such as sleep macrostructure [5,8,9,11,13,16,22,23,28,61,70], sleep microstructure [12], dream contents [59,60,62], spatial perception task [29–31,56], and spatial memory tasks [36,37]. A representative sample size for each category is shown as following:

1. Sleep architecture and circadian: A previous study investigated the differences in the power spectral activity between early and late blind in the different sleep stages, considering circadian desynchronization as a covariate. According to the power analysis for the ANCOVA, with $\alpha = 0.05$ and power $= 0.80$, a total of 22 participants (11 early blind and 11 late blind) are needed to achieve a significant result if a medium effect size (Cohen's $f = 0.897$) is expected, as indicated by the generalized eta squared ($\eta g^2 = 0.446$) to detect differences in delta activity in the N3 stage [12].

2. Dream visual contents: A study investigated the significant differences in the percentage of dreams with visual content between congenital blind (CB), late blind (LB), and sighted controls (SC). Based on a Kruskal-Wallis test with $\eta^2 = 0.429$, $\alpha = 0.05$, and desired power $= 0.80$, a total sample size of 18 participants (6 for each group) is required across the three groups (CB, LB, SC) to detect a significant difference in visual dream content [60].

3. Spatial perception task: The perception task that we will perform during this protocol study was previously evaluated between early blind and sighted controls [56]. According to the power analysis for the two-way ANOVA, with $\alpha = 0.05$ and power $= 0.80$, a total of 18 participants (9 blind and 9 sighted) are needed to achieve a significant result if a medium effect size (Cohen's $f = 0.577$) is expected, as indicated by the generalized eta squared ($\eta g^2 = 0.25$) to detect a difference in the spatial perception task.

4. Spatial memory task: The memory task we will perform during this study was previously evaluated between participants with visual deficits and sighted controls [37]. According to the power analysis for the ANOVA, with $\alpha = 0.05$ and power $= 0.80$, a total of 13 participants (6 blind and 7 sighted) are needed to achieve a significant result if a large effect size (Cohen's $f = 0.85$) is expected, as indicated by the generalized eta squared ($\eta g^2 = 0.419$) to detect a difference in the memory span.

Based on these statistical analyses of the sample size, a number of 10 blind and 10 sighted participants should be sufficient to detect significant and reliable differences between groups on different variables. This agrees with most of the sleep studies in blind participants that showed a sample number of 11/12 for the experimental and control group [7,8,11,12,16,29–31,36,37,56,59]. However, given the interest in investigating the onset of blindness, the high risk of withdrawal among blind participants due to the complexity of the protocol, and the proof-of-concept nature of the study exploring the interrelationship between various measures, we provisionally considered a sample size of 40 participants (20 BSI and 20 S). Nonetheless, efforts will be made to conduct interim analyses to estimate effect sizes based on the primary outcome, allowing for adjustments to the initial assumptions as necessary.

## Inclusion and exclusion criteria

Inclusion criteria for the S group also include normal or corrected-to-normal vision, while for the BSI group, an impairment of the peripheral visual system (i.e., involving pre-chiasmatic structures, such as the retina and optic nerve). The visual deficit can be congenital (from birth) or have a late onset. Participants with visual impairment, classified according to the current diagnostic criteria, must have residual vision lower than 1.3 LogMAR. Inclusion criteria for both groups of participants will be as follows: age between 18 and 85 years; any gender and sex; any ethnicity, provided they have a

good knowledge of the Italian language. Participants will be excluded from the study if they meet any of the following conditions: Tactile hypersensitivity (specifically, the ability to tolerate the equipment will be assessed); Hearing impairments; Use or history of using neuroactive drugs or substances in the six months prior to the study; Clinically significant and uncontrolled medical conditions (also diagnosed sleep disorders); Central nervous system disorders, including a history of seizures or convulsive episodes of any kind, even minor; Uncontrolled cardiopulmonary conditions that could affect sleep macro- or microstructure; Pregnant women. IQ scores below the normal threshold according to a recognized international scale.

### Recruitment and screening

The clinical investigation commenced with the enrollment of the first patient on April 4th, 2024, and the recruitment period is scheduled to conclude in October 2025. An extension of the recruitment period will be considered if the required number of participants is not reached.

Potential participants have been informed about the study via the Italian Institute of Technology mailing list, to which they have previously consented to be registered, or through communications with collaboration centers. All participants have been recruited at the IRCCS Ospedale Policlinico San Martino. The individuals of the control group have been recruited following the recruitment of the first experimental group and have been selected based on their characteristics (biological sex and age). Participants have been reimbursed for expenses/lost earnings related to traveling to the study site at the end of the protocol, with a maximum value of 75 euros (divided into 3 phases corresponding to the number of times they need to reach the venue). Before initiating data collection, participants have received comprehensive study information and the consent form. Inclusion and exclusion criteria have been evaluated verbally in an interview and by considering demographics during this screening stage. If all criteria have been met, participants have been deemed eligible for the study.

### Criteria for withdrawal/discontinuation of participation

The participant involvement in the study will end under the following conditions: 1) at their request to discontinue early; 2) if logistical or technical issues arise; 3) if they no longer meet the inclusion/exclusion criteria; 4) or at the discretion of the experimenter. Throughout the study, a subject's continued participation is contingent upon their adherence to the inclusion and exclusion criteria. If these criteria change, the experimenter will reassess the subject's participation, and the study may be discontinued to ensure the subject's safety and well-being, as well as the integrity of the data. Data collected up until that point will be retained and analyzed. The participant may withdraw from participation at any time without being required to provide a justification.

### Design and procedure

This study, titled "The impact of years of blindness on sleep and dream processes and the relationship with spatial abilities", was approved by the Liguria Regional Ethics Committee (623/2022 – DB id 12600) on 11th March 2024 and registered on ClinicalTrials.gov (ID NCT06631807).

The study design involves three experimental phases, with a total duration of 7 or 8 days per participant, as shown in Fig 1. The protocol was designed for 8 days to avoid overwhelming the participant with tasks on the first night. However, if the participant finds it more comfortable and simpler, all the equipment provided in visit 2 will be already directly provided on visit 1, shortening the protocol to 7 days without reducing the information collected. In that case, we collect both PSG and actigraphy data simultaneously during the first night.

The three phases include three different visits at the IRCCS San Martino Hospital sleep lab, during which various measurements will be assessed. All the technologies used in the protocol are summarized in Fig 2.

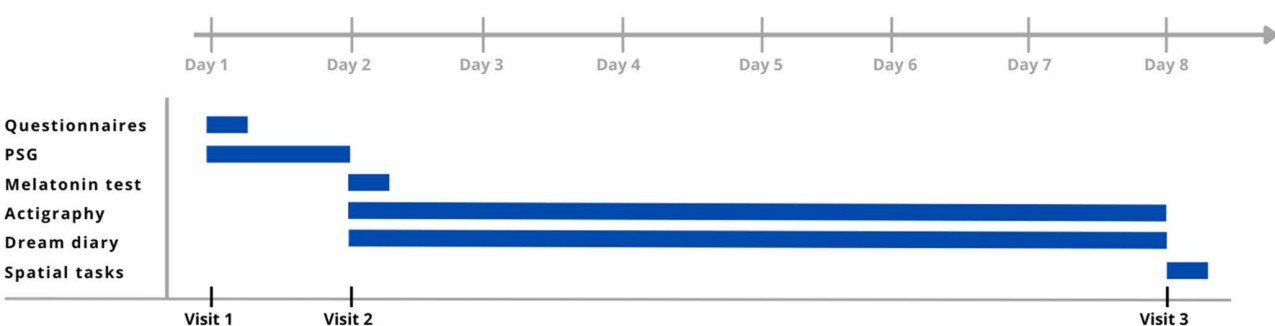

**Fig 1. Study design.** Figure shows the Gantt chart of the protocol.

**A.**

| Phase | Technology | Model | Assessment |
|---|---|---|---|
| 1 | PSG | Compumedics Somtè PSG2 | Sleep architecture and microstructure |
| 2 | Actigraphy | Act-Trust Condor Instrument | Circadian rhythm |
| 2 | Melatonin | Sleep Check Kit; Buhlmann | Circadian rhythm |
| 2 | Recorder | TASCAM DR-05 Linear | Dream diary |
| 3 | Speakers | - | Spatiotemporal audio bisection task |
| 3 | Audio-Corsi | - | Audio-Corsi task |

**B.**

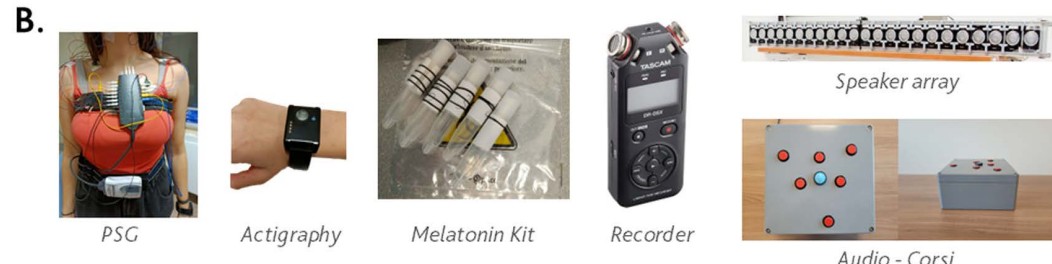

PSG   Actigraphy   Melatonin Kit   Recorder   Speaker array   Audio - Corsi

**Fig 2. Devices used in the protocol study.** 2.A. Table of technologies used for each phase. 2.B. Picture of the technologies.

## Phase and visit 1: screening and sleep and dream assessment

The protocol will start with collecting written consent and demographic data after reading the consent document to the participant. If the participant confirms to take part in the study the protocol will start.

Visit 1 defines the $t = 0$ of the protocol and takes place in the second half of the afternoon starting from 4.00 pm and accordingly with participant needs. It includes the measurements of sleep and dream quality using questionnaires, and sleep architecture by home-polysomnography (PSG) for one night. The questionnaires will evaluate sleep quality through the Pittsburgh Sleep Quality Index (PSQI) [71,72], Morningness-Eveningness Questionnaire (MEQ) [73], and Pre-Screening Questionnaire to Predict N24HSWD in Blind Subjects [74], and dream-related aspects using the Dream Recall Frequency Scale (DRFS) [75], and Van Dream Anxiety Scale (VDAS) [76]. The experimenter will read the questionnaires

and collect the answers verbally. We will also collect information on the sensory content of their dreams using a customized questionnaire based on previous studies [59,60]. Full polysomnography will then be set up using home polysomnograph Compumedics Somtè PSG2, conducted according to current international standards of the AASM [77], with the following channels: EEG (F3, F4, C3, C4, O1, O2), two monopolar EOG channels, three submental EMG channels (two recording electrodes and one reference), nasal cannula for oronasal airflow, peripheral pulse oximeter, and thoraco-abdominal movement bands. Once it is set up, the participant will remain seated for a few minutes at rest, and they may be asked to perform tasks to measure background brain activity, such as closing and opening the eyes. Afterward, they will be discharged with the home polysomnograph and instructed to return the following day to remove the device.

## Phase and visit 2: circadian and dream assessment

Visit 2 will take place the day after Visit 1 in the morning, around 8:00 am. The phase 2 will begin with the removal of the polysomnograph. Afterward, the participant will be fitted with an actigraphy bracelet, which they must wear for the following week. The actigraphy device (Act-Trust Condor Instrument) records the body movement, skin, ambient temperature, and ambient light.

On the first day of recording, the participants will be asked to collect five consecutive saliva samples, one hour apart in the evening, at home, for melatonin concentration analysis (Sleep Check Kit; Buhlmann). The samples will be collected independently by the participant, at one-hour intervals, so that the fourth sample (t = 0) coincides with their typical bedtime (−3, −2, −1, 0, and +1). For the fifth sample, participants are free to choose whether to stay awake one hour longer that evening or to go to sleep at their usual time and set an alarm to wake up and collect the last sample. The five test tubes for sample collection will be provided with a number of rubber bands on the edge corresponding to the number of the test tube to be considered. Therefore, at timepoint 1, test tube n 1 will have one rubber band; at timepoint 2, test tube n 2 will have two rubber bands, and so on. This has proven to be a simple and effective method for blind people to recognize the correct test tube. The participant will also be asked to keep a sleep and dream diary throughout the week. For this, they will be asked to verbally record information about the sleep and dream activity from the previous night. To collect subjective sleep information, participants are asked to report their bed and wake times, the quality of their sleep the previous night, any nighttime awakenings, and any naps taken during the day. About dreams activity, participants are asked to record any dream-related experiences immediately upon waking. A dream is defined broadly, not just vivid or complex "movie-like" scenarios, but any conscious experience during sleep. For this reason, participants were invited to record everything that was going through their mind just before waking. The focus should be on the final moments before waking, capturing perceptions, thoughts, emotions, or actions. If the participant remembers a dream, they should describe it. If they feel they dreamt but can't recall details, they should note that impression. If they remember nothing, they should state that clearly.

## Phase and visit 3: neuropsychological assessment

Visit 3 will be done one week after providing the actigraphic bracelet and recorder and will take place in the first half of the afternoon, starting between 2:00 pm to 4:00 pm. This phase evaluates the spatial perceptual and mnemonic functions in a session lasting approximately two hours maximum. The tests will be assessed at the same time of the day in all participants to avoid circadian variations. The neuropsychological tests will include the spatiotemporal acoustic bisection task [56] and the audio-corsi test [37]. In the first task, participants will be asked to judge the spatial or temporal relationship between three acoustic stimuli provided by an array of speakers (as shown in Fig 2). For spatial memory assessment, the experiments will use the equipment for the audio-corsi test [37,78]. The participant will listen to a series of white noise sounds moving along a predefined path in space. The participant's task will be to remember the spatial sequence of the different sounds. In both tests, the stimuli are controlled by a laptop with dedicated Matlab software that allows us to define stimulation parameters and record responses.

## Data management plan

Data management will be conducted in accordance with the General Data Protection Regulation (GDPR) and with the requirements of the Declarations of Helsinki [79]. Each participant will be assigned a general alphanumerical code (e.g., S001) at the beginning of the protocol. Personal-identifiable data will be kept separately from the experimental data, and the code to decrypt the personal data will only be available to authorized personnel.

Experimental data will be collected using a paper-based Case Report Form (CRF) in a pseudonymized format, which will be subsequently transferred to the Foundation for entry into a secure database hosted on a protected cloud platform. This database will be accessible exclusively to authorized personnel, with a tracking of all modifications. Security measures are robust, ensuring that pseudonymized data will be stored in duplicates in the institute's internal repository, thereby limiting access to authorized staff only. To further bolster data protection, comprehensive training on data protection principles will be provided to all staff involved, alongside established protocols for managing potential data breaches. Informed consent and collected data obtained from each participant will be securely archived for ten years, ensuring that access to personal data remains restricted to authorized staff.

## Data analysis plan

### Hypotheses

The first hypothesis aims to investigate the relationship between circadian disorders in blind participants and potential alterations in the behavioral tasks, given the high prevalence of N24SWD in blind individuals [2,3], with its occurrence being dependent on the degree of vision loss [6], and the impact that circadian misalignment can have on daily performance [25–28].

Specifically, H1.1. hypothesizes that visual deprivation causes a desynchronization of the circadian rhythm in congenitally BSI individuals and those who experience late-onset blindness after many years of visual deprivation. Previous evidence shows that blind people with a late onset, but many years of blindness are more likely to resemble the congenital or early-onset blind in their performance [56]. H1.2. hypothesizes that greater circadian desynchronization correlates with poorer performance on spatial perception and memory tasks in blind individuals.

The circadian rhythm disruptions could also affect the sleep architecture. However, it is not completely clear whether the alterations in sleep quality, architecture, and microstructure in blind individuals are due to plastic changes in brain structures induced by visual deprivation or only to circadian misalignment [7–13,16]. Therefore, H2.1. hypothesizes that circadian desynchronization affects sleep quality and sleep micro- and macrostructure in BSI individuals. Since there is a possibility that circadian desynchronization may not be the only factor influencing sleep micro- and macrostructure but also differences in underlying thalamo-cortical and cortico-cortical networks could still be reflected in sleep features, H2.2. hypothesizes that a correlation exists also between sleep macro and microstructure and spatial perceptual and memory performance. Indeed, we propose that BSI participants with a normally modulated circadian rhythm by the light-dark cycle may exhibit significant differences compared to controls, involving sleep patterns related to sensorimotor processing and the consolidation of spatial information.

Finally, it was reported that the sensory content of dreams is also affected by blindness [57,58], with visual dream content inversely correlated with the onset of blindness [59,60]. Given a demonstrated difference in some cognitive performances between high dream recallers (those who recall dreams more than three times per week) and low dream recallers, such as creativity, attention, and visuomotor tasks [80–82] H3.1. hypothesizes that a relationship between the frequency of dream recall and spatial tasks in both groups exists. Specifically, higher frequency of dream recall predicts better spatial abilities. Moreover, H3.2. hypothesizes that a greater presence of visual dream content can be associated with better spatial performance, suggesting whether sensory dream contents can influence waking abilities. We also propose that a higher frequency of eye movement may predict better spatial skills in blind individuals. Indeed, some theories

suggest an association between the visual reproduction of images in dreams and eye movement ( often reduced in blind individuals), which reflect the dreamer's gaze during dream imagery and the activation of the visual cortex [83–86].

## Data analysis

For addressing the hypotheses presented in this protocol different parameters will be considered for each questionnaire, assessing device and behavioral task.

The scores of questionnaires that will be considered in the analysis include: the PSQI global score and its 7 subcomponents that evaluate the subjective sleep quality, latency, duration, efficiency, sleep disorders, use of sleep medication and daytime dysfunction. Then, the analysis will include the MEQ global score, the Pre-Screening Questionnaire to Predict N24HSWD global score only for blind participants, and the VDAS global score. For the DRF scale, we will consider both the subjective, based on participant's answer, and objective, based on dreaming during the protocol week, score. Based on the score we will also classify participants as high or low dream recaller. Finally, an evaluation of the sharpness of the five senses in their dreams will be evaluated in a scale from 0 to 5. The scoring of dreams reports begins with the transcription of audio recordings, which are first preprocessed to remove contextual elements unrelated to the dream. Subsequently, the reports will be analyzed selecting the sensory category keywords to assess the sensory content of the dreams, following procedures used in previous studies [60,62,87].

From PSG data we will focus on EEG macro- and microstructure, and eye movements signals. The other polygraphic channels will be used to score sleep stages using 30-s epochs by two expert technicians, in accordance with the American Academy of Sleep Medicine manual criteria [77]. Sleep macrostructure parameters will include bedtime, wake up time, total sleep time (TST) in minutes and the percentage of stages N1, N2, N3, and REM within the TST, sleep latency onset (SL), sleep efficiency (SE), REM latency, number of awakenings, and wake after sleep onset (WASO). We will also explore the delta (0.5–4.5 Hz) and sigma (10–16 Hz) spectral activities, measured separately in N1, N2, N3 and REM sleep stages. For the analysis of sleep microstructure, we will detect automatically sleep spindles and slow waves with a visual inspection of the expert technicians to extract different features. Spindle features will include the mean spindle frequency (Hz), density (#/min), duration (s), amplitude (µV), power (dB) calculating the event-related spectral perturbation (ERSP) time-locked to the spindle onset, considering all the frequencies from 0.5 to 32 Hz. Separate analyses will be conducted for spindles detected in the frontal and central areas, as well as for slow (10–13 Hz) and fast (13–16 Hz) spindles. Slow waves features will include density (#/min), duration (s), amplitude of the maximum negative-peak (µV), down-slope (µV/s), up-slope (µV/s), topographical changes and spectral analysis. Slow oscillations-spindle coupling will be also explored.

From actigraphy data we will extract the described macrostructural parameters averaged on the whole week, such as bedtime, wake up time, time in bed, TST, SL, SE, number of awakenings, and WASO. Using the activity data record during the entire week, the actogram will be calculated to estimate the phase shift [88]. The circadian rhythm will be also assessed with cosinor analysis to extract the mesor (midline estimating statistic of rhythm), amplitude, and acrophase [89]. Non-parametric circadian parameters will be also calculated in terms of intradaily stability and variability. The circadian rhythm will be also assessed with the melatonin analysis calculating the DLMO (Dim Light Melatonin Onset).

Finally, two different tasks will be conducted at the end of the protocol. The audio temporal and spatial bisection task will explore the perceptual performance in the temporal and spatial domains. The performance of this psychophysical task will be evaluated in terms of Point of Subjective Equivalence (PSE) and Just Noticeable Difference (JND) [29,56]. Specifically, we will calculate the proportion of trials where the second sound is perceived as closer to the third sound and data will be fitted by cumulative Gaussian functions. Following standard psychophysical procedure, PSE and threshold estimates will be obtained from the mean and standard deviation of the best fitting function. The Audio-Corsi is a task that evaluates the working memory spatial abilities in terms of memory span as previously performed [37,78].

## Statistical analysis

Statistical analysis for investigating all hypotheses will involve parametric and non-parametric tests, with group differences assessed using t-tests, ANOVA, and linear or generalized linear mixed models as applicable. Both between-group and within-group correlations are of interest. In cases of statistical significance, an appropriate post-hoc test will be performed. Where necessary, results will be corrected for multiple comparisons. A significance level of $p < 0.05$ will be applied. Standard software will include MATLAB and R [69], which are widely recognized in research contexts. The EEGlab [90], FieldTrip [91], and custom Matlab scripts will be used to analyze electroencephalographic data.

Specifically, H1.1 and H1.2 will be tested using data from actigraphy and melatonin samples correlated to clinical data and performance in the audio bisection (i.e., JND and PSE) and Audio-Corsi tasks (i.e., memory span). The circadian data will include the phase shift calculated from actogram, the variables extracted from cosinor analysis, the non-parametric circadian parameters and the DLMO.

For the sleep quality and quantity analysis and their relationship with circadian rhythm (H2), we will also consider PSG data and questionnaires (global PSQI score and subcomponents, global MEQ score, and global Pre-Screening for N24HSWD for participants). From PSG and actigraphy data, we will extract general macrostructural sleep statistics (e.g., TST, SE, SL, etc.) to assess overall sleep quality and macrostructure, along with spectral analysis derived from EEG data. Then, from EEG data, we will also analyze sleep microstructure extracting the spindles and slow waves features. For H2.1. the circadian phase and other circadian parameters will be considered a covariate in the models, as performed in previous articles [12,16]. Control and BSI groups will also be compared for H2.1. and H2.2.

Finally, for the assessment of dream role (H3), the association between dream questionnaires (DRF, VDAS, dream sensory contents scale) or sensory contents information extracted from dream reports and spatial performance in the perceptual and memory tasks will be evaluated considering both within and between group analysis. Similar association analyses will be performed considering the eye movement activity [59].

## Discussion and limitations

This study addresses the relationship between different aspects of sleep and spatial performance in blind individuals. Indeed, little is known about how blindness might influence the relationship between sleep, including dreams, and spatial cognition. Could circadian disruption be the primary factor behind most of the sleep structure differences between blind and sighted individuals, or are other mechanisms involved? How does information reprocessing change compared to a person with typical vision, and how might this affect spatial performance? Is the key found in dreams or in neurophysiological biomarkers?

This study seeks to fill these gaps by examining the connections between clinical, psychological, neurobiological, and electrophysiological factors, offering a thorough understanding of how blindness impacts these interrelated areas. Moreover, the results could guide the development of new rehabilitative strategies or tools tailored to the needs of blind individuals, enhancing their quality of life and cognitive functioning. Indeed, different studies suggest that sensory stimulation during sleep, such as auditory or vibrotactile cues, may enhance the activation of newly learned memories, potentially improving spatial performance [66–68].

However, some limitations persist regarding sample size and the capacity to fully characterize visual impairment. Nonetheless, given the challenges of recruiting participants for a week-long observational study, significant findings may still emerge. We will try to account for the characterization of factors defining visual impairment. Additional limitations relate to the applicability of microstructural analyses with low-density EEG; nonetheless, we aim to apply more advanced analyses than those typically found in the literature in blind individuals, better characterizing the sleep neurophysiology and the actigraphy profile. Despite these constraints, we believe the findings from this study will contribute substantial insights and advancements to research in this field.

## Author contributions

**Conceptualization:** Helene Vitali, Dario Arnaldi.

**Data curation:** Laura Giorgetti, Francesco Famà.

**Formal analysis:** Helene Vitali, Claudio Campus, Pietro Mattioli.

**Funding acquisition:** Monica Gori.

**Investigation:** Laura Giorgetti, Francesco Famà.

**Methodology:** Helene Vitali, Claudio Campus, Laura Giorgetti, Francesco Famà, Pietro Mattioli.

**Project administration:** Helene Vitali, Dario Arnaldi, Monica Gori.

**Resources:** Dario Arnaldi, Monica Gori.

**Software:** Laura Giorgetti, Francesco Famà, Pietro Mattioli.

**Supervision:** Claudio Campus, Dario Arnaldi, Monica Gori.

**Writing – original draft:** Helene Vitali.

**Writing – review & editing:** Claudio Campus, Laura Giorgetti, Francesco Famà, Pietro Mattioli, Dario Arnaldi, Monica Gori.

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
