## [Decision Letter · Decision Letter 0]

Dear Dr. Vitali,

We look forward to receiving your revised manuscript.

Kind regards,

Serena Scarpelli

Academic Editor

PLOS ONE

Journal Requirements:

2.  Thank you for stating in your Funding Statement: [This research was partially supported by the European Research Council (PI Monica Gori; Grant agreement No. 948349)].

3. Thank you for stating the following in the Acknowledgments Section of your manuscript: [This research was partially supported by the European Research Council (PI Monica Gori; Grant agreement No. 948349)]

Please remove any funding-related text from the manuscript and let us know how you would like to update your Funding Statement. Currently, your Funding Statement reads as follows: “This research was partially supported by the European Research Council (PI Monica Gori; Grant agreement No. 948349).

https://erc.europa.eu/homepage

No the funder plays any role in the study design, data collection and analysis, decision to publish, or preparation of the manuscript.”

5. Please update your submission to use the PLOS LaTeX template. The template and more information on our requirements for LaTeX submissions can be found at http://journals.plos.org/plosone/s/latex .

Reviewers' comments:

Reviewer's Responses to Questions

**Comments to the Author**

1. Does the manuscript provide a valid rationale for the proposed study, with clearly identified and justified research questions?

Reviewer #1: Partly

Reviewer #2: Yes

2. Is the protocol technically sound and planned in a manner that will lead to a meaningful outcome and allow testing the stated hypotheses?

Reviewer #1: Partly

Reviewer #2: Yes

3. Is the methodology feasible and described in sufficient detail to allow the work to be replicable?

Reviewer #1: No

Reviewer #2: Yes

4. Have the authors described where all data underlying the findings will be made available when the study is complete?

Reviewer #1: Yes

Reviewer #2: No

5. Is the manuscript presented in an intelligible fashion and written in standard English?

Reviewer #1: Yes

Reviewer #2: Yes

You may also provide optional suggestions and comments to authors that they might find helpful in planning their study.

Reviewer #1: The research protocol aims to investigate the impact of blindness on sleep processes and analyze how sleep processes, dream content can affect spatial abilities.

The protocol is very interested and promising; however, it has some critical issues:

1) The introduction should be implemented with a more precise theoretical background consistent with the protocol assumptions. In particular, I suggest:

1.1) Describing the types of alterations in micro- and macrostructural sleep patterns observed in blind people and in those with rhythm disturbances (lines 56-59).

1.2) Describing how spatial perception is characterized in blind people and explaining how the results of Lockley and colleagues' 2008 study inform the hypotheses of the present experimental protocol (lines 65-70). I suggest a meta-analysis to expand the literature background. Bleau, M., Paré, S., Chebat, D. R., Kupers, R., Nemargut, J. P., & Ptito, M. (2022). Neural substrates of spatial processing and navigation in blindness: An activation likelihood estimation meta-analysis. Frontiers in neuroscience, 16, 1010354. https://doi.org/10.3389/fnins.2022.1010354

1.3) Expand on the section from lines 80 to 83. In particular, I suggest clarifying what is meant by: 'Can the varied processing of dream content influence how sensory information is processed during our waking experiences, significantly affecting long-term performance, such as spatial abilities?' Additionally, please indicate the relevant theory. Are you also referring to mnemonic consolidation processes?"

The current literature on dreaming suggests that waking experiences can be incorporated into dream content and reprocessed. Some studies have observed that sensory stimulation in sleep affects dream content. Additionally, some theories speculate that dreaming may play a role in sleep-dependent memory consolidation processes.

The following are some relevant reference articles.

-Wamsley EJ. Dreaming and offline memory consolidation. Curr Neurol Neurosci Rep. 2014 Mar;14(3):433. doi: 10.1007/s11910-013-0433-5. PMID: 24477388; PMCID: PMC4704085

-Scarpelli S, Alfonsi V, Gorgoni M, De Gennaro L. What about dreams? State of the art and open questions. J Sleep Res. 2022 Aug;31(4):e13609. doi: 10.1111/jsr.13609. Epub 2022 Apr 13. PMID: 35417930; PMCID: PMC9539486.

-Bloxham, A., & Horton, C. L. (2024). Enhancing and advancing the understanding and study of dreaming and memory consolidation: Reflections, challenges, theoretical clarity, and methodological considerations. Consciousness and cognition, 123, 103719. https://doi.org/10.1016/j.concog.2024.103719

2) Method.

Inclusion and exclusion criteria

2.1) In lines 152-153, when discussing medical conditions, are you also referring to other sleep disorders? If so, please specify.

Screening and Recruitment

2.2) For screening and recruitment, specify whether test batteries/interviews are administered to participants and which ones.

Design and Procedures:

2.3) At line 188 which task are you referring to? In case it is not clear how the information collection is not reduced. How do you restructure the protocol? Are PSG and actigraphy recorded simultaneously, or is one less day of actigraphy recorded?

Visit 2

2.4) Explain the instructions for dream collection. Are dreams collected in the morning within 15 minutes of waking up?

2.5) During Visit 2 why are sleep diaries not also provided to collect subjective information?

Date analysis

Hypothesis

2.5) Regarding hypotheses, it is necessary to specify what kind of spatial ability is tested in your protocol. Additionally, some hypotheses are unclear:

2.5.1) What do you mean that circadian desynchronization affects sleep structure? Are you referring in terms of micro- and macrostructure? When you talk about "other factors" in line 285, what factors are you referring to?

2.5.2) Hypothesis H2.2 suggests that there may be a correlation between spatial performance and macro- and micro-structure if circadian desynchronization is not the only factor impacting night sleep in the blind. However, so formulated it is not properly correct. it is possible that both desynchronization and other sleep alterations may affect performance tasks.

2.5.3) Hypothesis H3.1 is not entirely accurate if you are referring to dream frequency. The literature does not suggest that dream frequency directly impacts performance on specific tasks. Instead, it indicates that waking elements might be incorporated into dreams and reprocessed, but the role in memory processes is not yet clear. Hypothesis H3.2 would be more accurate if tasks were administered before and after a night of sleep to assess whether task incorporations correlate with improved performance the following morning.

Statistical analysis

2.6) Given the amount of collected measures, I suggest implementing a dedicated paragraph “data analysis” to describe the computation of different variables. For example: For example:

(a) PSG data:How will sleep scoring be conducted, and which variables will be extracted? Will spectral analyses be performed? Will spectral analyses be performed? For spindle and slow wave analyses, will a detection be performed and what parameters will be used?

(b) Regarding actigraphic measures, how will they be computed and what measures will be extracted?

(c) Melatonin measures: What analysis processes will be used?

(d) What measures will be used for performance tasks?

(e) What kind of scoring is done for dreams?

(f) Questionnaire measures: Which measures will be included in the statistical analyses? For example, will the PSQI global score or its subcomponents be used?

2.7) Following this, I then suggest restructuring the statistical analyses section to specify the analyses that will be carried out for the different hypotheses, indicating for each analysis the variables considered.

3) Refernces

In the text the some references is not numbered: For example: line 68, lines 271-272, lines 339-340, line 342

Reviewer #2: Dear Authors,

Thank you for the opportunity to review your manuscript. Overall, I find the research questions to be interesting, clearly defined, and well-grounded in theory. The study protocol is well-designed and methodologically sound. I have only minor comments that may help improve the reporting of the protocol in the manuscript. Please, see the attachment for detailed comments.

Recruitment Process:

Please clarify why certain eligibility criteria are not checked before scheduling the first visit. Would this not be part of the recruitment process?

Terminology and Wording Adjustments:

Consider replacing 'quantity' with a more precise term like 'architecture', as you are measuring more than just sleep duration.

The term 'protocol' may be more appropriate than 'sleep measurements' since your study includes questionnaires as well.

'Dream quality' is somewhat ambiguous; please clarify what is meant by this term.

Study Procedure:

It would be helpful to specify whether participants arrive at the hospital in the evening or at another time.

If the fourth sample coincides with their typical sleep time, does this mean they must delay sleep for the fifth sample? Please clarify the instructions given to participants.

If participants arrive in the morning, please explicitly state this in the manuscript.

Hypotheses:

In the hypotheses section, rephrase them as declarative statements rather than questions. For example, instead of posing a question, state that 'Visual deprivation causes a desynchronization of the circadian rhythm in...'

Discussion:

The discussion section seems to contain mostly the same information as in the introduction, which doesn't seem that helpful to me.

Minor revisions improve the reporting of the protocol the manuscript's clarity and precision. I appreciate the effort that has gone into this study and believe it contributes valuable insights to the field. I look forward to seeing the final version.

Best regards,

**Do you want your identity to be public for this peer review?** For information about this choice, including consent withdrawal, please see our Privacy Policy

Reviewer #1: No

Reviewer #2: **Yes: ** Emma Peters

---

## [Author Response · Author response to Decision Letter 1]

20 May 2025

We would like to thank Editor and the Reviewers for the time and effort in reviewing our manuscript and for the insightful comments proposed. We have carefully examined the points raised and addressed them in the revised version of our manuscript. Here a copy of the response rebuttal letter uploaded in this revision.

Journal Requirements:

The PLOS ONE style templates can be found at

We reviewed PLOS ONE's style requirements and revised our manuscript accordingly. We also uploaded figures separately as individual files.

2. Thank you for stating in your Funding Statement: [This research was partially supported by the European Research Council (PI Monica Gori; Grant agreement No. 948349)].

We added the sentence “There was no additional external funding received for this study” following the previous Funding Statement. We also provided an amended Funding Statement within your cover letter.

3. Thank you for stating the following in the Acknowledgments Section of your manuscript: [This research was partially supported by the European Research Council (PI Monica Gori; Grant agreement No. 948349)]

Please remove any funding-related text from the manuscript and let us know how you would like to update your Funding Statement. Currently, your Funding Statement reads as follows: “This research was partially supported by the European Research Council (PI Monica Gori; Grant agreement No. 948349). https://erc.europa.eu/homepage No the funder plays any role in the study design, data collection and analysis, decision to publish, or preparation of the manuscript.”

We removed any funding-related text from the manuscript and updated the Funding Statement as follow “This research was partially supported by the European Research Council (PI Monica Gori; Grant agreement No. 948349). https://erc.europa.eu/homepage No the funder plays any role in the study design, data collection and analysis, decision to publish, or preparation of the manuscript. There was no additional external funding received for this study.”

We also provided an amended Funding Statement within your cover letter.

This manuscript refers to a study protocol that is already publicly available on ClinicalTrials.gov (ID: NCT06631807). At this stage, no data is yet available, as the study is still ongoing. Once data collection is completed and the data has been processed and analyzed, we will make the full dataset freely available through Zenodo, in line with PLOS ONE's open data policy.

5. Please update your submission to use the PLOS LaTeX template. The template and more information on our requirements for LaTeX submissions can be found at http://journals.plos.org/plosone/s/latex.

We prepared the manuscript using a Word file in accordance with PLOS ONE's style requirements. We will provide a version using the PLOS LaTeX template if required for the final submission.

Reviewers' comments:

Reviewer #1

The research protocol aims to investigate the impact of blindness on sleep processes and analyze how sleep processes, dream content can affect spatial abilities.

The protocol is very interested and promising; however, it has some critical issues:

The Reviewer correctly summarized the main aim of the protocol. We thank you very much for your precious feedback. We are convinced that, following your suggestions, we could address all the points you raised, clarify all the aspects you highlighted, and improve the scientific rigor of the manuscript. Below is a point-by-point response to each question raised. When we specified the Lines we referred to the track changes version of the manuscript.

1) The introduction should be implemented with a more precise theoretical background consistent with the protocol assumptions. In particular, I suggest:

1.1) Describing the types of alterations in micro- and macrostructural sleep patterns observed in blind people and in those with rhythm disturbances (lines 56-59).

Thank you for your suggestion. We have provided a more detailed description of the types of alterations in micro- and macrostructural sleep patterns in blind people and how circadian rhythm can influence them, see Lines 64-75.

1.2) Describing how spatial perception is characterized in blind people and explaining how the results of Lockley and colleagues' 2008 study inform the hypotheses of the present experimental protocol (lines 65-70). I suggest a meta-analysis to expand the literature background. Bleau, M., Paré, S., Chebat, D. R., Kupers, R., Nemargut, J. P., & Ptito, M. (2022). Neural substrates of spatial processing and navigation in blindness: An activation likelihood estimation meta-analysis. Frontiers in neuroscience, 16, 1010354. https://doi.org/10.3389/fnins.2022.1010354

Thank you for your suggestion. We moved part of the Discussion describing how spatial perception is characterized in blind people in the Introduction, as suggested by Reviewer 2 (Lines 88-104). We also explained the results found by Lockley and colleagues' 2008, see Lines 80-85. Finally, we also added the interesting meta-analysis suggested.

1.3) Expand on the section from lines 80 to 83. In particular, I suggest clarifying what is meant by: 'Can the varied processing of dream content influence how sensory information is processed during our waking experiences, significantly affecting long-term performance, such as spatial abilities?' Additionally, please indicate the relevant theory. Are you also referring to mnemonic consolidation processes?"

The current literature on dreaming suggests that waking experiences can be incorporated into dream content and reprocessed. Some studies have observed that sensory stimulation in sleep affects dream content. Additionally, some theories speculate that dreaming may play a role in sleep-dependent memory consolidation processes.

The following are some relevant reference articles.

-Wamsley EJ. Dreaming and offline memory consolidation. Curr Neurol Neurosci Rep. 2014 Mar;14(3):433. doi: 10.1007/s11910-013-0433-5. PMID: 24477388; PMCID: PMC4704085

-Scarpelli S, Alfonsi V, Gorgoni M, De Gennaro L. What about dreams? State of the art and open questions. J Sleep Res. 2022 Aug;31(4):e13609. doi: 10.1111/jsr.13609. Epub 2022 Apr 13. PMID: 35417930; PMCID: PMC9539486.

-Bloxham, A., & Horton, C. L. (2024). Enhancing and advancing the understanding and study of dreaming and memory consolidation: Reflections, challenges, theoretical clarity, and methodological considerations. Consciousness and cognition, 123, 103719. https://doi.org/10.1016/j.concog.2024.103719

Thank you for your suggestion. We expand the section on dreams in the Introduction part, clarifying also the relevant theory as suggested, see Lines 117-129.

2) Method.

Inclusion and exclusion criteria

2.1) In lines 152-153, when discussing medical conditions, are you also referring to other sleep disorders? If so, please specify.

We are generally asked if they suffered of clinically significant and uncontrolled medical in general, also including OSAS or already diagnosed sleep disorders. We specified it.

Screening and Recruitment

2.2) For screening and recruitment, specify whether test batteries/interviews are administered to participants and which ones.

For screening and recruitment, we only evaluated during an interview the inclusion and exclusion criteria specified in the previous session. We tried to better specify it.

Design and Procedures:

2.3) At line 188 which task are you referring to? In case it is not clear how the information collection is not reduced. How do you restructure the protocol? Are PSG and actigraphy recorded simultaneously, or is one less day of actigraphy recorded?

We considered restructuring the protocol to collect both PSG and actigraphy data simultaneously during the first night. We specified it in the manuscript. However, we decided not to adopt this as the baseline approach in order to avoid overwhelming participants with too much information on the first day. However, if participants demonstrate confidence and find the procedures easy to follow, we may offer this option as it shortens the overall protocol duration and is easier to integrate into routine clinical practice.

Visit 2

2.4) Explain the instructions for dream collection. Are dreams collected in the morning within 15 minutes of waking up?

2.5) During Visit 2 why are sleep diaries not also provided to collect subjective information?

Thank you for your suggestions. We added more details about how we collect dream information. We also specify that we collect subjective information about sleep. We do not provide any additional material to participants, but we ask them to report their bed and wake times, the quality of their sleep the previous night, any nighttime awakenings, and any naps taken during the day, see Lines 291-300.

Date analysis

Hypothesis

2.6) Regarding hypotheses, it is necessary to specify what kind of spatial ability is tested in your protocol. Additionally, some hypotheses are unclear:

2.6.1) What do you mean that circadian desynchronization affects sleep structure? Are you referring in terms of micro- and macrostructure? When you talk about "other factors" in line 285, what factors are you referring to?

As you correctly pointed out, we refer to sleep structure in terms of micro- and macrostructure. Moreover, as now specified in the Introduction, it remains unclear whether changes in sleep macro- and microstructure are a direct result of blindness or a consequence of circadian rhythm desynchronization. While accounting for circadian misalignment in analyses may appear to eliminate macro- and microstructural alterations, differences in underlying thalamo-cortical and cortico-cortical networks could still be reflected in sleep patterns. This is what we refer to as "other factors." These points have been clarified in the manuscript, see Lines 357-362.

2.6.2) Hypothesis H2.2 suggests that there may be a correlation between spatial performance and macro- and micro-structure if circadian desynchronization is not the only factor impacting night sleep in the blind. However, so formulated it is not properly correct. it is possible that both desynchronization and other sleep alterations may affect performance tasks.

Thank you for pointing out this imprecision. We agree with your comment. Indeed, as presented in H.1.2, we hypothesized that circadian desynchronization has an effect. What we aim to hypothesize in H.2.2 is that, even in the absence of circadian misalignment, differences in sleep micro- and macrostructure may still be observed due to mechanisms linked to changes in thalamo-cortical and cortico-cortical networks in the absence of vision. We have attempted to clarify this in the manuscript.

2.6.3) Hypothesis H3.1 is not entirely accurate if you are referring to dream frequency. The literature does not suggest that dream frequency directly impacts performance on specific tasks. Instead, it indicates that waking elements might be incorporated into dreams and reprocessed, but the role in memory processes is not yet clear.

Hypothesis H3.2 would be more accurate if tasks were administered before and after a night of sleep to assess whether task incorporations correlate with improved performance the following morning.

About Hypothesis H3.1, some studies have shown differences in certain cognitive performances, such as creativity, attention, and visuomotor tasks, between high dream recallers (those who recall dreams more than three times per week) and low dream recallers (Dumel et al., 2015; Ruby et al., 2022; Vallat et al., 2022), as well as some differences in the resting brain activity (Eichenlaub et al., 2014a; 2014b). The underlying mechanism is surely that dreaming may allow for greater reprocessing of information acquired during wakefulness, which in turn may influence the development of broader cognitive functions. Based on this evidence, we formulated Hypothesis H3.1. We have now clarified this point in the manuscript, see Lines 369-375.

Here, the references:

• Dumel, G., Carr, M., Marquis, L.-P., Blanchette-Carrière, C., Paquette, T. and Nielsen, T. (2015), Infrequent dream recall associated with low performance but high overnight improvement on mirror-tracing. J Sleep Res, 24: 372-382. https://doi.org/10.1111/jsr.12286

• Vallat R, Türker B, Nicolas A, Ruby P. High Dream Recall Frequency is Associated with Increased Creativity and Default Mode Network Connectivity. Nat Sci Sleep. 2022;14:265-275

https://doi.org/10.2147/NSS.S342137

• Perrine Ruby, Rémy Masson, Benoit Chatard, Roxane Hoyer, Laure Bottemanne, Raphael Vallat, Aurélie Bidet-Caulet, High dream recall frequency is associated with an increase of both bottom-up and top-down attentional processes, Cerebral Cortex, Volume 32, Issue 17, 1 September 2022, Pages 3752–3762, https://doi.org/10.1093/cercor/bhab445

• Jean-Baptiste Eichenlaub, Olivier Bertrand, Dominique Morlet, Perrine Ruby, Brain Reactivity Differentiates Subjects with High and Low Dream Recall Frequencies during Both Sleep and Wakefulness, Cerebral Cortex, Volume 24, Issue 5, May 2014, Pages 1206–1215, https://doi.org/10.1093/cercor/bhs388

• Eichenlaub, JB., Nicolas, A., Daltrozzo, J. et al. Resting Brain Activity Varies with Dream Recall Frequency Between Subjects. Neuropsychopharmacol 39, 1594–1602 (2014). https://doi.org/10.1038/npp.2014.6

About Hypothesis H3.2, we agree that a task-sleep-task design experiment could be more appropriate, and future studies will certainly move in that direction. However, to avoid further increasing the burden on participants within this protocol, we chose not to include pre- and post-sleep task assessments. However, we have revised the manuscript by replacing the term predict, which would indeed be inappropriate given our current design, with association. Our aim is to explore whether sensory content in dreams, potentially reflecting the reprocessing of daytime experiences, is generally associated wi

---

## [Decision Letter · Decision Letter 1]

Investigating the Impact of the Years of Blindness on Sleep Rhythms, Dream Patterns, and Spatial Abilities: The BLINDREAM Protocol

PONE-D-25-01572R1

Dear Dr. Vitali,

We’re pleased to inform you that your manuscript has been judged scientifically suitable for publication and will be formally accepted for publication once it meets all outstanding technical requirements.

Kind regards,

Serena Scarpelli

Academic Editor

PLOS ONE

Additional Editor Comments (optional):

Reviewers' comments:

Reviewer's Responses to Questions

**Comments to the Author**

1. Does the manuscript provide a valid rationale for the proposed study, with clearly identified and justified research questions?

Reviewer #1: Yes

Reviewer #2: Yes

2. Is the protocol technically sound and planned in a manner that will lead to a meaningful outcome and allow testing the stated hypotheses?

Reviewer #1: Yes

Reviewer #2: Yes

3. Is the methodology feasible and described in sufficient detail to allow the work to be replicable?

Reviewer #1: Yes

Reviewer #2: Yes

4. Have the authors described where all data underlying the findings will be made available when the study is complete?

Reviewer #1: No

Reviewer #2: Yes

5. Is the manuscript presented in an intelligible fashion and written in standard English?

Reviewer #1: Yes

Reviewer #2: Yes

You may also provide optional suggestions and comments to authors that they might find helpful in planning their study.

Reviewer #1: The manuscript has improved significantly with the revisions. The authors have comprehensively responded to all comments, and I therefore recommend acceptance

Reviewer #2: Dear Authors,

Thank you for addressing each of my comments. I accept this revision and believe the authors appropriate incorporated my suggestions and requests and at this stage, I recommend accepting this manuscript for publication.

**Do you want your identity to be public for this peer review?** For information about this choice, including consent withdrawal, please see our Privacy Policy

Reviewer #1: No

Reviewer #2: **Yes: ** Emma Peters

---

## [Editor Report · Acceptance letter]

PONE-D-25-01572R1

PLOS ONE

Dear Dr. Vitali,

I'm pleased to inform you that your manuscript has been deemed suitable for publication in PLOS ONE. Congratulations! Your manuscript is now being handed over to our production team.

Kind regards,

on behalf of

Dr. Serena Scarpelli

Academic Editor

PLOS ONE